# Isolation of Two New Compounds and Other Constituents from Leaves of *Piper crocatum* and Study of Their Soluble Epoxide Hydrolase Activities

**DOI:** 10.3390/molecules24030489

**Published:** 2019-01-30

**Authors:** Hong Xu Li, Seo Young Yang, Young Ho Kim, Wei Li

**Affiliations:** 1College of Pharmacy, Chungnam National University, Daejeon 34134, Korea; charon0077@gmail.com; 2Korean Medicine (KM) Application Center, Korea Institute of Oriental Medicine, Daegu 41062, Korea

**Keywords:** phenolic, *Piper crocatum*, Piperaceae, soluble epoxide hydrolase activity

## Abstract

Two new phenolic glucosides, pipercroside A and B (**1** and **2**), along with 10 known compounds were isolated from the leaves of *Piper crocatum* Ruiz & Pav. Their chemical structures were elucidated through extensive spectroscopic analyses, including 1D and 2D NMR experiments and HR-ESI-MS analysis and comparison with previously reported data. All the isolated compounds were assessed for soluble epoxide hydrolase (sEH) inhibitory activity. Among them, erigeside II (**5**) showed inhibitory activity with an IC_50_ value of 58.5 µM.

## 1. Introduction

The genus *Piper* belongs to the Piperaceae family and comprises around 700 species distributed across both hemispheres [1]. Piper plants have possessed high commercial and medicinal significance for thousands of years, with spices being the most valued specimens. Apart from being widely used as common food adjuncts as flavoring, seasoning, coloring agents, and preservatives, spices are also known to have a variety of antioxidant effects and properties [2]. In this regard, many plants are used for therapeutic purposes as exceptional sources of phytochemicals such as phenolic compounds [3]. Moreover, phenolic compounds are the most widely distributed secondary metabolites, arising biogenetically from either the shikimate/phenylpropanoid pathway or deriving phenylpropanoids [4], and fulfill a broad range of physiological roles. For instance, many studies have disclosed that a large number of phenolic compounds obtained from various species of the Piperaceae family showed cytotoxicity and antifungal potential [5], and their applicability as an important source of antiprotozoal/antimicrobial agents has been also suggested [6]. Indeed, the antioxidant activity of phenolic compounds in higher plants has long been known [7]. However, to the best of our knowledge, chemical investigations on the constituents of *Piper crocatum* are still scarce, and the presence of phenolic compounds and sesquiterpenes in its extracts have not been reported. Soluble epoxide hydrolase (sEH) is widely distributed in mammalian tissue, with potent effects on the biological activities conducted by the cardiovascular and urinary systems [8]. It is responsible for the hydrolysis of epoxyeicosatrienoic acids (EETs), which are endothelium-derived hyperpolarizing factors (EDHFs) that act as regulators of vascular function [9]. The sEH can convert EETs to their corresponding diols (dihydroxyeicosatrienoic acids, DHETs), and reduce the EETs’ effects on cardiovascular systems through vasodilation, antimigration of vascular smooth muscle cells, and anti-inflammatory action. Therefore, sEH was considered as a potential therapeutic target for vascular disease [10]. 

## 2. Results and Discussion

The studies outlined above were interesting in identifying the constituents of *P. crocatum* responsible for its therapeutic activity. In this work, 12 compounds were isolated from the MeOH extract of *P. crocatum* (Figure 1). Their structures were elucidated through 1D and 2D NMR spectroscopy and mass spectrometry analyses and identified as pipercroside A (**1**), pipercroside B (**2**), 2,5-dimethoxy-3-glucopyranosylcinnamic alcohol (**3**) [11], cimidahurinin (**4**) [12], erigeside II (**5**) [13], syringin (**6**) [14], β-phenylethyl β-d-glucoside (**7**) [15], methyl salicylate 2-*O*-β-d-glucopyranoside (**8**) [16], icariside D1 (**9**) [17], 4-hydroxybenzoic acid β-d-glucosyl ester (**10**) [18], benzyl β-d-glucoside (**11**) [17], and phenylmethyl 6-*O*-α-l-arabinofuranosyl-β-d-glucopyranoside (**12**) [19].

Compound **1** was isolated as a colourless gum. Its molecular formula was established as C_17_H_26_O_9_ using HR-ESI-MS. The ^1^H-NMR spectrum of **1** (Table 1) exhibited a singlet assignable to a symmetrical 1,3,4,5-tetrasubstituted aromatic ring at δ_H_ 6.56 (H-2/H-6), two methoxy signals at δ_H_ 3.84 (s, 3-OCH_3_ and 5-OCH_3_), and a signal attributable to the anomeric proton of the glucosyl moiety at δ_H_ 4.80 (dd, *J* = 7.6, 1.0 Hz, H-1′) was consistent with the β-configuration of the glucose. Meanwhile, the ^13^C-NMR spectrum exhibited six aromatic carbons at δ_C_ 108.3 (C-2/C-6), 137.5 (C-1), 154.0 (C-3/C-5), and δ_C_ 134.7 (C-4) and signals attributable to the glucosyl moiety at δ_C_ 62.4 (C-6′), 71.2 (C-4′), 75.7 (C-2′), 77.8 (C-3′), 78.3 (C-5′), and 105.5 (C-1′). The presence of a 2-propanol moiety that gave rise to signals at δ_C_ 46.7 (C-7), 69.7 (C-8), and 23.1 (C-9) was elucidated with the help of ^1^H–^1^H correlation spectroscopy (COSY) and heteronuclear multiple quantum correlation (HMQC) spectrum. Further, the corresponding heteronuclear multiple bond correlation (HMBC) spectrum confirmed the planar structure through the following correlation peaks: H-1′ (δ_H_ 4.80)/C-4 (δ_C_ 134.7), H-7 (δ_H_ 2.64 and 2.70)/C-1 (δ_C_ 137.5), C-2/6 (δ_C_ 108.3), and the two methoxy groups H-3/5-OCH_3_ (δ_H_ 3.84)/C-3/5 (δ_C_ 108.3). (Figure 2 and Appendix A) The absolute configuration of **1** was determined through optical rotation and rotating frame nuclear overhauser effect spectroscopy (ROESY) spectrum. Thus, the ROESY spectrum suggested a correlation between H-7a (δ_H_ 2.70) and H-8 (δ_H_ 3.96) that, along with a large coupling constant between both protons of 7.0 Hz, permitted the establishment of the β-orientation of H-8. Moreover, the optical rotation of **1** was found to be −64.8. By comparing with the reported optical rotation values for (*R*)-3-phenylpropan-2-ol and (*S*)-1-phenylpropan-2-ol, which were −24.3 and +50.8, respectively [20,21], the configuration of C-8 was determined to be *R*. Therefore, compound **1** was established to be (8*R*)-8-(4-hydroxy-3,5-dimethoxy) propan-8-ol-4-*O*-β-d-glucopyranoside and named pipercroside A.

Compound **2** was also isolated as a colourless gum. Its molecular formula was established to be C_17_H_24_O_9_ through HR-ESI-MS. The ^1^H-NMR spectrum of **2** was very similar to that of known compound **5**, except for the integration for the aromatic protons (**2** has one and **5** has two aromatic protons), indicating that the glucosyl moieties of both compounds were identical. The ^13^C spectrum of **2** exhibited six aromatic carbons at δ_C_ 111.2 (C-2), 123.8 (C-1), 139.2 (C-4), 142.4 (C-5), 143.8 (C-6), and 146.7 (C-3), two methoxy groups at δ_C_ 57.8 (3-OCH_3_) and 61.9 (5-OCH_3_), and allyl and glucosyl signals that were similar to those of **5**. The planar structure of **2** disclosed in Figure 1 was further validated by the corresponding HMBC spectrum through the following cross peaks: H-7 (δ_H_ 3.33)/C-2 (δ_C_ 111.2), C-9 (δ_C_ 115.7), C-1 (δ_C_ 123.8), C-8 (δ_C_ 138.2), C-6 (δ_C_ 143.8), H-3-OCH_3_ (δ_H_ 3.78)/C-3 (δ_C_ 146.7), and H-5-OCH_3_ (δ_H_ 3.87)/C-3 (δ_C_ 142.4). The absence of a correlation between the carbon atoms of the methoxy groups with H-7 suggested that the methoxy groups were substituted symmetrically as in **5**. The deshielding of C-2 (δ_C_ 111.2) could be explained by the presence of a hydroxy group at the meta-position. Taken together, these data permitted the establishment of the structure of **2** as 1-allyl-3,5-dimethoxy-6-hydroxy-4-β-d-glucopyranoside, which was named pipercroside B.

Overall, 12 phenolic compounds were isolated in this study from the MeOH extract of the leaves of *P. crocatum*. Besides the new compounds, to the best of our knowledge, this paper constitutes the first report on the isolation of compounds **3**–**5**, **8**–**10**, and **12** from the genus of *Piper* and **6**, **7** and **11** from *Piper crocatum*.

Finally, the soluble epoxide hydrolase (sEH) inhibitory effect were investigated of the isolated phenolic glucosides (**1**–**12**) from *P. crocatum*. The sEH enzyme is widely distributed in mammalian tissue, with potent effects on the on the biological activities conducted by the cardiovascular and urinary systems [8]. It is responsible for the hydrolysis of epoxyeicosatrienoic acids (EETs), which are endothelium-derived hyperpolarizing factors (EDHFs) that act as regulators of vascular function [9]. Thus, sEH can convert EETs to their corresponding diols (dihydroxyeicosatrienoic acids, DHETs), thereby reducing their effects on the cardiovascular system through vasodilation, antimigration of vascular smooth muscle cells, and anti-inflammatory action. Therefore, sEH has been considered as a potential therapeutic target for vascular disease [10].

The sEH inhibitory activities were determined using recombinant human sEH incubated with PHOME, an artificial substrate for fluorescence detection (Table 2).

The results revealed that only **4**, **5**, **6**, and **8** exhibited inhibitory effect, with **5** having the highest sEH inhibitory activity (92.9%). Compounds **4**, **6**, and **8** showed weak sEH inhibitory activities of 8.8%, 17.4%, and 18.9%, respectively. Interestingly, compound **2**, despite having a structure that is similar to that of **5**, exhibited no inhibitory effect, which might be due to the presence of the –OH group at the C-6 position affecting the steric hindrance of the allyl moiety, which resulted in a decrease of the sEH inhibitory effect. Similarly, the difference in the inhibitory effect of the isomeric compounds **3** and **6** could stem from the presence of the methoxy group at the meta-position in **3**, eliminating the sEH inhibitory effect. Although **8** and **10** can be considered similar compounds, **10** revealed no inhibitory effect, whereas **8** exhibited low sEH inhibitory effect. This suggests that the chemical circumstance of the ketone moiety in **8** (located at the ortho-position) and **10** (connected with the glucose moiety) was significant for the inhibitory effect. Finally, a comparison of all the compounds indicated that the allyl group or the double-bond moiety might contribute to the sEH inhibitory effect. The IC_50_ value found for **5** (58.5 µM) demonstrates the potential of the phenolic derivatives isolated from *P. crocatum* as bioactive compounds.

## 3. Materials and Methods

### 3.1. General Information

Optical rotations were determined using a Jasco DIP-370 automatic polarimeter (Jasco, Tokyo, Japan Manufacturer, City, State if USA/Canada, Country). The NMR spectra were recorded using a BRUKER AVANCE III 600 (^1^H, 600 MHz; ^13^C, 150 MHz) (Bruker Biospin GmbH, Karlsruhe, Germany), with tetramethylsilane (TMS) as an internal standard. Heteronuclear multiple quantum correlation (HMQC), heteronuclear multiple bond correlation (HMBC), rotating frame nuclear overhauser effect spectroscopy (ROESY), and ^1^H–^1^H correlation spectroscopy (COSY) spectra were recorded using a pulsed field gradient. The HR-ESI-MS spectra were obtained by using an Aglient 1200 LC-MSD Trap spectrometer (Agilent, Santa Clara, CA, USA). Preparative HPLC was performed using a GILSON 321 pump, 151 UV/VIS detector (Gilson, VILLIERS-LE-BEL, France), and RStech HECTOR-M C_18_ column (5-micron, 250 × 21.2 mm) (RS Tech Crop, Chungju, South Korea). Column chromatography was performed using a silica gel (Kieselgel 60, 70–230, and 230–400 mesh, Merck, Darmstadt, Germany), YMC C-18 resins, and thin layer chromatography (TLC) was performed using pre-coated silica-gel 60 F_254_ and RP-18 F_254S_ plates (both 0.25 mm, Merck, Darmstadt, Germany), the spots were detected under UV light and using 10% H_2_SO_4_.

### 3.2. Plant Material

Dried leaves of *Piper crocatum* Ruiz & Pav were collected from Cilendek Timur, Bogor, West Java, Indonesia, in August 2016 and taxonomically identified (Identification number: 1714/IPH.1.01/If.07/VIII/2016) by the staff at herbarium Laboratory, Research Center for Biology, Indoenesian Institute of Sciences, Cibinong, West Java, Indonesia. A voucher specimen (BBRC-SMPLS-003) was deposited at the Herbarium of Aretha Medika Utama, Biomolecular and Biomedical Research Center, Bandung, West Java, Indonesia.

### 3.3. Extraction and Isolation

The dried leaves of *P. crocatum* (2.6 kg) were refluxing extracted with MeOH (5 L × 3 times). The total extraction (400.0 g) of MeOH was suspended in deionized water and partitioned with Hexane, and water fraction. Then the water fraction was partitioned sequentially with EtOAc and BuOH, yielding EtOAc (1A, 16.1 g) and BuOH (1B, 65.0 g). The EtOAc fraction was subjected to a silica gel column chromatography with a gradient of CHCl_3_-MeOH-H_2_O (10:1:0, 9:1:0, 8:1:0, 6:1:0.1, 5:1:0.1, 4:1:0.1, 3:1:0.1, 2:1:0.1, MeOH 2.0 L for each step) to give 11 fractions (Fr. 1A-1–1A-11). Fractions 7 and 8 were combined and isolated with a gradient of MeOH-Water (1:2, 1:1 and MeOH) by MPLC using YMC C_18_ column to give 8 fractions (Fr. 2A-1–2A-8). Subfraction 2A-8 was separated using a silica gel column chromatography with a CHCl_3_-MeOH (20:1) elution solvent to give compound 5 (6.0 mg). Fraction 9 was isolated with a gradient of MeOH-Water (1:2, 1:1 and MeOH) by MPLC using YMC C_18_ column to give one fraction, and then it was isolated by prep-HPLC to give compounds 11 (3.0 mg). The BuOH fraction was subjected to a silica gel column chromatography with a gradient of CHCl_3_-MeOH-H_2_O (10:1:0, 9:1:0, 8:1:0, 6:1:0.1, 5:1:0.1, 4:1:0.1, 3:1:0.1, 2:1:0.1, MeOH 5.0 L for each step) to give 11 fractions (Fr. 2B-1–2B-11). Fractions 2B-2 and 2B-3 were combined and isolated with a gradient of MeOH-H_2_O (1:2, 1:1 and MeOH) by MPLC using YMC C_18_ column to give 12 fractions (Fr. 1C-1–1C-12). Fraction. 1C-3 was separated using a Sephadex LH-20 column and eluted by MeOH and its sub fraction was isolated by prep-HPLC to give compound 6 (3.0 mg). Fraction. 1C-7 was separated using a Sephadex LH-20 column and eluted by MeOH and its sub fractions were isolated by prep-HPLC to give compound 8 (4.9 mg). Fraction. 1C-8 was separated using a Sephadex LH-20 column and eluted by MeOH and its sub fraction was isolated by prep-HPLC to give compound 7 (3.6 mg). Fraction. 1C-9 was separated using a Sephadex LH-20 column and eluted by MeOH and its sub fraction was isolated by prep-HPLC to give compound 2 (2.0 mg). Fractions 2B-4 and 2B-5 were combined and isolated with a gradient of MeOH-H_2_O (1:2, 1:1 and MeOH) by MPLC using YMC C_18_ column to give 13 fractions (1D-1–1D-13). Fraction. 1D-2 was separated by a Sephadex LH-20 column and eluted by MeOH and its sub fraction was isolated by prep-HPLC to give compound 3 (15.0 mg). Fraction. 1D-3 was separated using a Sephadex LH-20 column and eluted by MeOH and its sub fraction was isolated by prep-HPLC to give compound 1 (3.0 mg). Fraction. 1D-6 was separated using a Sephadex LH-20 column and eluted by MeOH and its sub fraction was isolated by prep-HPLC to give compound 9 (30.4 mg). Fractions 2B-6 and 2B-7 were combined and isolated with a gradient of MeOH-H_2_O (1:2, 1:1 and MeOH) by MPLC using YMC C_18_ column to give 10 fractions (1E-1–1E-10). Fraction. 1E-2 was separated using a Sephadex LH-20 column and eluted by MeOH and its sub fraction was isolated by prep-HPLC to give compounds 4 (3.0 mg) and 10 (2.0 mg). Fraction. 1E-5 was separated using a Sephadex LH-20 column and eluted by MeOH and its sub fraction was isolated by prep-HPLC to give compound 12 (31.2 mg).

*Pipercroside A (***1***)*: Colorless gum; C_17_H_26_O_9_; $[\alpha {]}_{D}^{25}$: –64.8 (*c* 0.51, MeOH); ^1^H-NMR (MeOD, 600 MHz) and ^13^C-NMR data (MeOD, 150 MHz), see Table 1 HR-EI-MS *m*/*z* 397.1475 [M + Na]^+^ (calcd for C_17_H_26_O_9_Na, 397.1443)

*Pipercroside B (***2***)*: Colorless gum; C_17_H_24_O_9_; $[\alpha {]}_{D}^{25}$: –5.2 (*c* 0.05, MeOH); ^1^H-NMR (MeOD, 600 MHz) and ^13^C-NMR data (MeOD, 150 MHz), see Table 1 HR-EI-MS *m*/*z* 395.1318 [M + Na]^+^ (calcd for C_17_H_24_O_9_Na, 395.1281).

### 3.4. Soluble Epoxide Hydrolases (sEH) Activity Assay

The soluble epoxide hydrolases (sEH) inhibitory assay, Bis-Tris (B9754), and albumin (A8806) were purchased from Sigma Aldrich (St. Louis, MO, USA). The human recombinant soluble epoxide hydrolases (sEH, 10011669), PHOME (10009134) and 12-(3-adamantan-1-yl-ureido)-dodecanoic acid (AUDA) (10007972) were purchased from Caymanchem (Cayman, MI, USA). The 96-well white plate was purchased from Costar (CORNING, NY, USA). The fluorescence intensity measurement was conducted on using a Tecan infinite F200 microplate reader (Tecan, Mannedorf, Switzerland).

The enzymatic assays were performed according to the methods modified from previous research [10]. The 130 µL aliquot of recombinant human sEH (12.15 ng/mL) was diluted with buffer (25 mM Bis-Tris-HCl, pH 7.0 containing 0.1 mg/mL BSA) and mixed with 20 µL of MeOH, and then with 50 µL of 3-phenyl-cyano(6-methoxy-2-naphthalenyl) methyl ester-2-oxiraneacetic acid (PHOME, 10 µM) was added. The amount of product converted from substrate by the enzyme was measured by fluorescence photometry (excitation filter 330 nm, and emission filter 465 nm), as follows: Enzyme activity (%) = [S_40_ − S_0_/C_40_ − C_0_] × 100. Where C_40_ and S_40_ are the fluorescence of control and inhibitor after 40 min, and S_0_ and C_0_ are the fluorescence of inhibitor and control at 0 min. In this testing, AUDA was used as a positive control.

## 4. Conclusions

In conclusion, for the first time 12 phenolic compounds were isolated from the leaves of *Piper crocatum* Ruiz & Pav, including two new phenolic glucosides, pipercroside A and B. Their structures were elucidated through NMR and mass spectrometry, and their sEH inhibitory activity was evaluated to find that erigeside II (5) exhibited inhibitory activity, with an IC_50_ value of 58.5 µM.

## Figures and Tables

**Figure 1 molecules-24-00489-f001:**
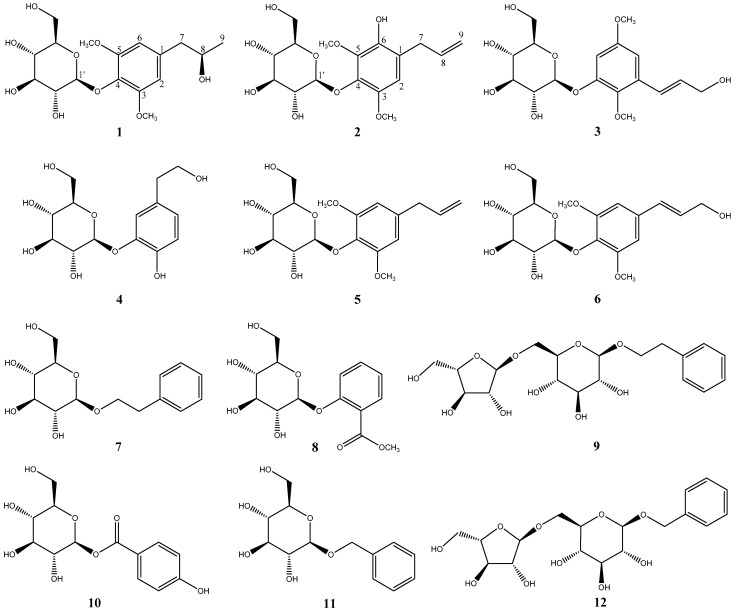
Structures of compounds **1**–**12** isolated from cultures of *P. crocatum*.

**Figure 2 molecules-24-00489-f002:**
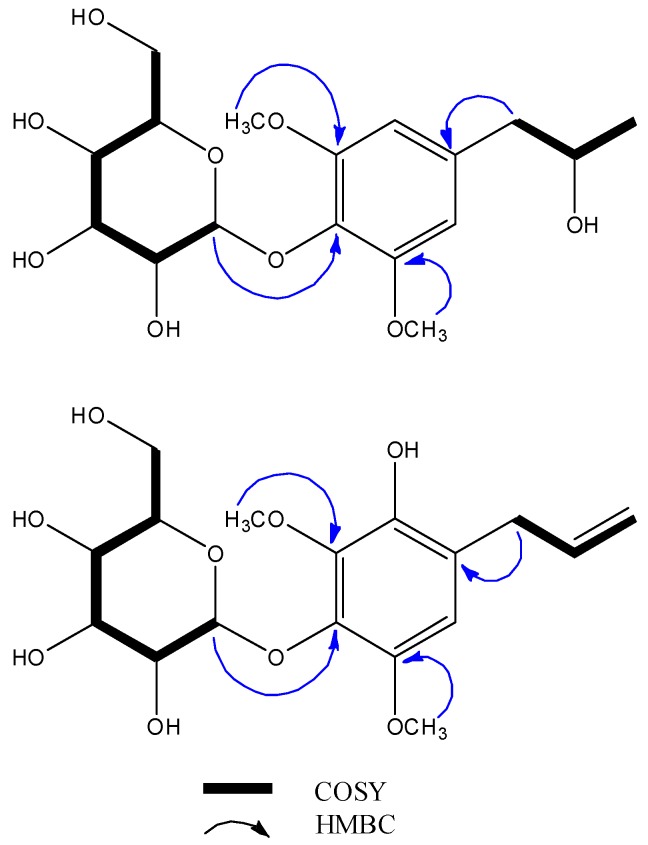
^1^H–^1^H correlation spectroscopy (COSY) and key heteronuclear multiple bond correlation (HMBC) correlations between compounds **1** and **2**.

**Table 1 molecules-24-00489-t001:** ^1^H-(600 MHz) and ^13^C-NMR (150 MHz) spectroscopic data of compounds **1** and **2** (MeOD, δ, ppm, *J*/Hz).

	1		2	
Pos.	δ_H_	δ_C_	δ_H_	δ_C_
1	-	137.5	-	123.8
2	6.56, s	108.3	6.53, s	111.2
3	-	154.0	-	146.7
4	-	134.7	-	139.2
5	-	154.0	-	142.4
6	6.56, s	108.3	-	143.8
7a	2.70, dd, 13.4, 7.0	46.7	3.33, m	35.1
7b	2.64, dd, 13.4, 6.0		-	
8	3.96, q, 6.0	69.7	5.95, dd, 17.0, 10.0, 6.6	138.2
9a	1.17, d, 6.0	23.1	5.05, dd, 17.0, 2.0	115.7
9b	-		5.00, dd, 10.0, 1.0	
1′	4.80, dd, 7.6, 1.0	105.5	4.99, d, 7.5	104.9
2′	3.47, td, 7.5, 2.4	75.7	3.46, m	75.7
3′	3.42, d, 2.8	77.8	3.41, m	77.8
4′	3.41, d, 2.6	71.2	3.41, m	71.3
5′	3.20, dq, 7.0, 2.5	78.3	3.21, dd, 8.1, 4.0	78.3
6′a	3.78, dd, 12.0, 2.3	62.4	3.78, dd, 12.0, 2.3	62.4
6′b	3.67, dd, 12.0, 5.1		3.66, dd, 12.0, 2.3	
3-OCH_3_	3.84, s	56.91	3.78, s	57.8
5-OCH_3_	3.84, s	56.91	3.87, s	61.9

**Table 2 molecules-24-00489-t002:** Inhibitory effects of isolated compounds **1**–**12**.

Compounds	100 µM (%)	IC_50_ (µM)
**1**	N.I ^c^	N.T ^a^
**2**	N.I	N.T
**3**	N.I	N.T
**4**	8.8 ± 2.5	N.T
**5**	92.9 ± 0.5	58.5 ± 0.5
**6**	17.4 ± 0.6	N.T
**7**	N.I	N.T
**8**	18.9 ± 1.5	N.T
**9**	N.I	N.T
**10**	N.I	N.T
**11**	N.I	N.T
**12**	N.I	N.T
**AUDA** ^b^		13.3 ± 0.8

sEH activity was expressed as the percentage of control activity. Values represent means ± SD (*n* = 3). ^a^ N.T: Not Tested. ^b^ 12-(3-adamantan-1-yl-ureido)-dodecanoic acid (AUDA) was used as the positive control. ^c^ N.I: Not Inhibition.

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
