# Peer review of "Isolation of Two New Compounds and Other Constituents from Leaves of Piper crocatum and Study of Their Soluble Epoxide Hydrolase Activities"

_molecules, 2019, doi:10.3390/molecules24030489_

Author Response

Answers of Reviewers' Comments

Manuscript ID: molecules-437865

Title: Isolation of two new compounds and other constituents from leaves of Piper crocatum and study of their soluble epoxide hydrolase activities

Corresponding Author: Dr. Wei Li

We greatly appreciate your kind consideration on our manuscript.
According to reviewers’ comments, several parts of the manuscript have been corrected.

Reviewers' comments to Author:

Discovery of new bioactive molecules from renewable natural plant resources is a very important issue for the R&D of drug leading compounds. In this work, the authors isolated two new phenolic glucosides, pipercroside A and B (1 and 2), along with 10 known compounds from the leaves of Piper crocatum Ruiz & Pav. Their chemical structures were elucidated by 1D and 2D NMR experiments and HR-ESI-MS analysis and comparison with previously reported data. By testing the soluble epoxide hydrolase (sEH) inhibitory activity, erigeside II (5) showed high inhibitory activity (IC50 = 58.5 μM) among the 18 isolated compounds. The work demonstrated the extract/isolation and molecular characterization methods of the natural compounds from the Piper crocatum Ruiz & Pav, and the sEH inhibition properties were preliminarily studied, providing useful information for the discovery of natural plant-based sEH inhibitors towards vascular disease treatment. There are some issues need to be addressed:

1. In general, the type and concentration of bioactive compounds in plants greatly rely on their harvested organs/regions such as leaf, peel, seed, root, etc, for establishing the optimized bioactive compound profiles of Piper crocatum Ruiz & Pav, why did the authors choose the leaves of P. Crocatum samples in priority for MeOH extraction?

Answer: For the folk prescription in Indonesia the leaves of Piper crocatum were widely used for the treatment for various disease, also the previous literatures report mainly focused on the extractions and fractions of the leaves for in-vivo and in-vitro tests. Therefor we choose the leaves of P. crocatum samples in priority for MeOH extraction.

2. It is noticed that the optical rotation value of the compound Pipercroside A (1): [α]D25: –64.8 (c 0.51, MeOH) and Pipercroside B(2):[α]D25:–5.2 (c 0.0005, MeOH) have~12-fold difference while their concentration have ~1000-fold difference, interestingly, these two compounds have very similar structures, for a better comparison, the authors please re-measure the[α]D25 value of Pipercroside A (1)under the same concentration(c 0.0005, MeOH)of Pipercroside B(2).

Answer: We are sorry for this kind of mistake, due to the mistyping these two compounds have 1000-fold of the concentration differences. The actual concentration for Pipercroside B(2) was (c 0.05, MeOH). Because of the minimal isolated amount, there was not much remaining sample after the bioassay tests and physical and chemical tests.

3. There are some errors and unclear parts in the manuscript, such as: “page1, line 39. with potent effects on the on the..”;  Moreover, the abbreviation of control compound AUDA should be given in the experimental part 12-(3-adamantan-1-yl-ureido)-dodecanoic acid (AUDA).Authors please check and carefully revise the manuscript.

Answer: This comment was corrected in the revised manuscript.

4. There are also many errors in the reference part, such as “ref13, Miyase, T.; Kuroyanagi, M.; Noro, T.; UENO,A.; FUKUSHIMA, S.J.C.; bulletin, p...”; “ref17MIYASE, T.; UENO, A.; TAKIZAWA, N.; KOBAYASHI, H.; OGUCHI, H.J.C.;…”and so on, Authors please carefully check and revise the reference part.

Answer: This comment was corrected in the revised manuscript.

Reviewer 2 Report

Dear Authors,

a clear presentation and prepared well as a communication; only a view points I suggest:

- avoid "we" in the description as this not scientific sounding

- L133; italics style of strain designations; correct throughout the manuscript,

- I would suggest to describe the sEH assay a bit more clear and what actually happens there - as it here it is just briefly described ... and basically this is important when describing a new compound.

Suppl.:

Figures S1 and S8 are not readable; the format need tob e optimized for presentation.

Author Response

Answers of Reviewers' Comments

Manuscript ID: molecules-437865

Title: Isolation of two new compounds and other constituents from leaves of Piper crocatum and study of their soluble epoxide hydrolase activities

Corresponding Author: Dr. Wei Li

We greatly appreciate your kind consideration on our manuscript.
According to reviewers’ comments, several parts of the manuscript have been corrected.

Reviewers' comments to Author:

Dear Authors,
a clear presentation and prepared well as a communication; only a viewpoint I suggest:

- avoid "we" in the description as this not scientific sounding

Answer: This comment was corrected in the revised manuscript.

- L133; italics style of strain designations; correct throughout the manuscript,

Answer: This comment was corrected in the revised manuscript.

- I would suggest describing the sEH assay a bit clearer and what actually happens there - as it here it is just briefly described ... and basically this is important when describing a new compound.

Answer: However, to our surprise, the new compounds we isolate exhibited with no sEH inhibition, due to some unclear parts of the sEH mechanism we were looking forward for some further study to indicate this problem.

Suppl.:
Figures S1 and S8 are not readable; the format needs to be optimized for presentation.

Answer: According to reviewer's comment, these figures were optimized in the supplementary data.
